# Epidemiology of Respiratory Syncytial Virus Hospitalizations in Poland: An Analysis from 2015 to 2023 Covering the Entire Polish Population of Children Aged under Five Years

**DOI:** 10.3390/v16050704

**Published:** 2024-04-29

**Authors:** Jan Mazela, Teresa Jackowska, Marcin Czech, Ewa Helwich, Oliver Martyn, Pawel Aleksiejuk, Anna Smaga, Joanna Glazewska, Jacek Wysocki

**Affiliations:** 1Department of Neonatology, Poznan University of Medical Sciences in Poznan, 60-535 Poznan, Poland; 2Department of Pediatrics, Centre of Postgraduate Medical Education, 01-813 Warsaw, Poland; tjackowska@cmkp.edu.pl; 3Department of Pharmacoeconomics, Hospital Infection Control Team, Institute of Mother and Child, 01-211 Warsaw, Poland; marcin.czech@imid.med.pl; 4Institute of Mother and Child, 01-211 Warsaw, Poland; ewa.helwich@imid.med.pl; 5Sanofi A/S, Vaccines Medical Affairs, DK-2100 København, Denmark; oliver.martyn@sanofi.com; 6Sanofi Sp. z o.o., Vaccines Medical Affairs, 01-211 Warsaw, Poland; 7PEX Sp. z o.o., 02-796 Warsaw, Poland; anna.smaga@pexps.pl (A.S.); joanna.glazewska@pexps.pl (J.G.); 8National Institute of Public Health NIH—National Research Institute, 00-791 Warsaw, Poland; 9Department of Preventive Medicine, Poznan University of Medical Sciences at Poznan, 61-701 Poznan, Poland; jwysocki@ump.edu.pl

**Keywords:** respiratory syncytial virus, RSV, acute respiratory tract infections, epidemiology, public health, Poland

## Abstract

Background: Respiratory syncytial virus (RSV) is an important cause of childhood hospitalizations. The aim of the study was to estimate the rates of RSV-related hospitalizations in children aged less than 5 years in Poland. Methods: This retrospective observational cohort study was based on data obtained from the National Health Fund in Poland regarding all acute respiratory tract infections and RSV-coded admissions of children (age < 5 years) to public hospitals between July 2015 and June 2023. Patients were stratified based on the following age groups: 0–1 month, 2–3 months, 4–6 months, 7–12 months, 13–24 months, and 25–60 months. Results: The number of RSV-related hospitalizations increased every season, both before and through the ending phase of the coronavirus disease 2019 (COVID-19) pandemic. The COVID-19 pandemic was associated with a shift in the seasonality pattern of RSV infection. Hospitalization rates per 1000 inhabitants were the highest for children aged 0–12 months, reaching 47.3 in the 2022/23 season. Within this group, the highest hospitalization rate was observed for children aged 2–3 months—94.9 in the 2022/23 season. During the ending phase of the COVID-19 pandemic, the observed increase in admission rates was 2-, 4-, and 5-fold the pre-COVID rate for children aged <12 months, 12–24 months, and 25–60 months, respectively. Conclusions: In Poland, RSV infections cause a significant burden in hospitalized children aged less than 5 years. RSV-related hospitalizations were most frequent in children aged less than 1 year. The COVID-19 pandemic was associated with a shift in the seasonality pattern of RSV infections. After the pandemic, more RSV-related hospitalizations were observed in older children (aged 13 months and older) vs. the pre-pandemic phase.

## 1. Introduction

Respiratory syncytial virus (RSV) causes a spectrum of acute respiratory tract infections (ARTIs), with severity ranging from mild upper respiratory symptoms to severe lower respiratory tract infections and outcomes [1]. RSV infections exhibit a seasonal and regional pattern, which typically peaks during the fall and winter months in temperate latitudes [2,3]. Understanding and predicting these patterns are essential for healthcare planning and the timely implementation of preventive measures [4].

RSV is a prominent global health concern, particularly due to its substantial impact on respiratory morbidity in vulnerable populations such as young children and older adults [5]. Young children, particularly infants and toddlers, are at increased risk of RSV infection due to factors such as an immature immune system, limited prior exposure to the virus, and close contact within daycare settings [6,7,8]. The clinical manifestations of RSV infections in this age group often include symptoms such as nasal congestion, cough, and fever; however, severe infections can lead to bronchiolitis and pneumonia. Preterm infants and those with underlying medical conditions, such as congenital heart disease or chronic lung disease, face an increased risk of developing inflammatory responses that can cause severe complications such as the exacerbation of asthma and an increased risk of recurrent wheezing. However, the majority (79%) of hospitalizations occur in healthy, full-term infants [9]. While most RSV infections in young children are self-limiting, severe cases may require hospitalization, posing a considerable burden on healthcare systems worldwide [1,10,11,12,13,14].

Globally, in 2019, an estimated 33 million RSV-related ARTIs led to 3.6 million hospital admissions and 101,400 overall deaths in children aged 5 years or younger [5]. An estimated 97% of RSV-caused deaths in children aged 0–60 months occurred in low- and middle-income countries [5]. Preventive strategies, such as the use of monoclonal antibodies for high-risk infants and a maternal vaccine to protect babies, are critical in mitigating the impact of RSV infections in young children [4,11,15].

In Poland, the incidence and disease burden of RSV infections among young children is not well understood and likely to be underestimated due to a previous lack of routine laboratory-based RSV testing and surveillance [15]. Since January 2023, the National Health Fund in Poland (Narodowy Fundusz Zdrowia [NFZ]) has covered rapid RSV tests, and since February 2023, physicians have been obliged to record RSV infections confirmed by laboratory testing [16].

Several new prevention methods are now available, including monoclonal antibodies and vaccines administered to pregnant women for the passive immunization of children. Additionally, comprehensive epidemiological data are essential to effectively implement these new preventive strategies [4,17]. Public health awareness campaigns, emphasizing hygiene practices and the importance of early medical intervention, also play a pivotal role in minimizing the spread of RSV among young children [18]. With these new preventative options on the horizon, a better understanding of the real-world RSV burden in all young children in Poland is critical to inform public health strategies [19]. Therefore, we retrospectively analyzed data from the NFZ in Poland to describe the disease burden of serious RSV infection among young children (<5 years) between July 2015 and June 2023 (eight consecutive seasons).

## 2. Methods

### 2.1. Study Design

This retrospective observational cohort study was based on anonymized data obtained from the NFZ in Poland regarding all ARTIs and RSV-coded admissions to public hospitals of children aged less than 5 years. NFZ data cover almost the entire Polish population, since nearly all patients in Poland are hospitalized in public hospitals. Data for the period between July 2015 and June 2023 were obtained, containing information about the patients’ age, sex, diagnosis, and comorbidities.

### 2.2. Definitions

International Classification of Diseases, Tenth Edition (ICD-10), codes used to identify ARTI hospitalizations, RSV-related hospitalizations, and children at high risk of severe RSV infection are shown in Appendix A. Risk factors for severe infection included prematurity, bronchopulmonary dysplasia, congenital malformations, Down syndrome, cystic fibrosis, and cerebral palsy.

### 2.3. Data Analysis

Patients were stratified based on the following age groups: 0–1 month, 2–3 months, 4–6 months, 7–12 months, 13–24 months, and 25–60 months. The number of hospital admissions was calculated for ‘respiratory years’ starting every year on July 1 and ending on June 30 of the following year, to capture the full annual epidemic each season.

To compute RSV-associated hospital admission rates per 1000 inhabitants, the number of RSV admissions in each age group was divided by the population size of this age group in the middle of each respiratory year (i.e., on 30 December). Reference data came from the Polish Central Statistical Office. The mean proportion of male live births from 2020 to 2023 was 51.3 ± 0.1%. For children aged less than 1 year, the population size of each group was estimated by dividing the total number of live births by 12, assuming an equal distribution of births in each month.

Due to the low number of cases, the 2020/21 season was excluded from the analysis of age impact. Data for the seasons between 2015/16 and 2019/20 were aggregated to calculate pre-coronavirus disease 2019 (COVID-19) values and data for the seasons 2021/22 and 2022/23 were aggregated to calculate post-COVID-19 values. For the purpose of this study, we considered seasons 2021/22 onwards as ‘post-COVID-19′ although it could be considered as the transition from a pandemic to an endemic phase. For both pre-COVID-19 and post-COVID-19 periods, the mean rates of hospitalizations were calculated as the number of admissions per 1000 inhabitants or 1000 live births. To compare changes in admission rates between age groups before and after the pandemic, the ratio of post- to pre-COVID-19 rates was calculated. Descriptive analyses of RSV epidemiology by epidemic season and age group were conducted.

Group comparisons were performed using two-way analysis of variance. Values of *p* less than 0.05 were considered statistically significant. All calculations were performed using the Statistica software (version 13, StatSoft, Kraków, Poland).

## 3. Results

### 3.1. Burden and Seasonality of RSV-Related Hospitalizations

The number of RSV-related hospitalizations increased every season, from 5202 in 2015/16 to 10,418 in the pre-pandemic 2019/20 season (Figure 1—blue bars). In the 2020/21 season, a very low number of cases (259) was observed, while in the two consecutive seasons, the number of hospitalizations was around twice as high as before the pandemic, reaching approx. 20,000. The upward trend observed from before the COVID-19 pandemic was noted again between the 2021/22 (10.8/1000) and 2022/23 (11.9/1000) seasons. There were fewer than five hospitalizations resulting in death in each season from 2015/16 to 2020/21, with the last two seasons recording seven and five deaths, respectively.

During the five seasons preceding the COVID-19 pandemic, the same seasonality pattern was observed, with the peak of hospitalizations in February and a significant number of cases recorded between December and April (Figure 2). After the 2020/21 season with almost no cases, a shift in seasonality appeared as the 2021/22 season started in summer 2021, achieved its peak in October, and ended in winter 2022. The timing of the 2022/23 season was later than the previous year with the peak of hospitalizations observed in December.

The share of RSV-related hospitalizations in all ARTI hospitalizations increased slightly every season, from 3.5% in 2015/16 to 15.1% in 2022/23, with the exception of 2020/2021 (Figure 3).

The rate of RSV-related hospitalizations per 1000 increased every season except 2020/21, ranging from 2.7 in 2015/16 to 11.9 in 2022/23. On the contrary, the rates of ARTI-related hospitalizations remained very similar for the first and last season (79.5 per 1000 in 2015/16 and 78.7 per 1000 in 2022/23; Figure 4).

Children in the risk groups accounted for 5.8% of all RSV-coded admissions (Table 1).

### 3.2. Number of Hospitalizations, by Age Group

The most frequent age group for RSV-coded hospitalizations was infants aged 2–3 months (30% before and 24% after the pandemic), followed by those aged 4–6 months (25% before and 22% after the pandemic). Data are shown in Figure 5. The share of children aged over 1 year, and especially those over 2 years, increased over the study period, while the share of infants aged 2–3 and 4–6 months decreased. Children aged less than 1 year accounted for 81% of all RSV-coded hospitalizations before the COVID-19 pandemic and for 70% of the hospitalizations after the pandemic.

Annual hospitalization rates per 1000 increased across seasons in all age groups. They were the highest for children aged 0–12 months: between 12.0 in the 2015/16 season and 47.3 in the 2022/23 season, and much lower for older children (Table 2). In the group of children aged less than 1 year, the highest hospitalization rates were observed for those aged 2–3 months: between 26.4 in the 2015/16 season and 94.9 in the 2022/23 season (Table 3). There was an over 5-fold increase in the admission rates before vs after the COVID-19 pandemic in children aged 25–60 months; over 3-fold, in those aged 12–24 months; and over 2-fold, in infants aged less than 1 year (*p* < 0.001) (Table 2).

## 4. Discussion

Our results showed a clear seasonality pattern of RSV-associated hospitalization rates in young children before the COVID-19 pandemic, with the highest hospitalization rates between December and April, peaking in February. This finding is in line with other publications suggesting that the peak of RSV infections worldwide occurs in the winter months [8,20,21,22,23,24,25,26].

This seasonal pattern was disrupted by the COVID-19 pandemic. We observed a very low number of RSV-related hospitalizations in young children during the 2020/21 season, followed by a significant increase during the next two seasons, with shifted peaks of RSV-associated hospitalizations in October 2021/22 and in December 2022/23. The lack of typical winter RSV outbreaks was also reported in studies from Italy [22], England [24], Germany [25], Sweden [27], and Australia [28]. This disruption in epidemiology has been attributed to public health interventions aimed at reducing the spread of COVID-19, including the closure of schools, daycare centers, and public meeting places as well as travel restrictions, physical distancing, and hygiene measures such as handwashing and wearing face masks [20,24]. Indeed, a temporal correlation was reported between COVID-19-related nonpharmaceutical interventions (NPIs) and the numbers of reported cases of other viral diseases [24,29,30]. These results suggest that COVID-19 preventive measures are linked to reduced RSV-related hospitalization rates in children.

The relaxation of public health interventions preceded RSV outbreaks that were shifted in time and intensity compared with previous seasons. According to our results, the 2021/22 serious RSV infection presentations among young children, i.e., hospitalizations, started in August, peaked in October, and ended around January. This is in line with observations from other studies [22,24,25,31,32]. We also noted that the subsequent 2022/23 season’s number of RSV-associated hospitalizations in young children peaked in December, which is closer to the pre-COVID-19 dynamics of serious RSV infections in that age group. Future studies should explore whether RSV seasonality returns to patterns seen before the COVID-19 pandemic or whether more variability in the timing of the season continues to be seen.

During the combined last two seasons of our analysis, which included the pandemic-ending period when NPIs were largely relaxed or ended, RSV-related hospitalization rates in different age groups were 2.4 to 5.6 times higher than before the COVID-19 pandemic. Similar findings were reported by Loconsole et al. [22], although the increase was smaller compared with our study. Another reason for the increased number of RSV-coded hospitalizations might be the increased availability and/or use of RSV infection diagnostic tests [33], especially during the COVID-19 pandemic. Moreover, RSV symptoms can be nonspecific and can overlap with those of influenza and COVID-19. Thus, empiric diagnosis can often be insufficient and should be supported by viral diagnostic testing to facilitate treatment and improve surveillance [34]. Therefore, it is possible that the use of RSV diagnostic tests significantly increased during the COVID-19 pandemic, more RSV-related admissions were recorded, and a bigger share of ARTI admissions could be coded as related to RSV—hence, a potential detection bias. This can also explain the increasing proportion of RSV-related hospitalizations among all ARTI admissions in our study. Stable rates of ARTI-related hospitalizations together with increasing rates of RSV-related hospitalizations between 2015/16 and 2022/23 also confirm this hypothesis. Other studies suggest that the pandemic may have led to changes in testing practices. In Germany, the use of viral testing before the pandemic was reported in only 8.7% of pediatric patients with symptoms of respiratory tract infection [35]. In the United States, the test volumes increased from 66,324 in the 38 months before the pandemic to 95,741 at 25 months after the pandemic [36]. Our results showed a slight increase in RSV-coded hospitalizations across seasons, both before and after the pandemic, which is in line with other publications [37,38,39,40]. 

In the last season of our analysis, RSV-coded admissions accounted for 15% of all ARTI hospitalizations. This is similar to data for other European countries, with the proportion ranging from 5% for England to 15% for Italy [19]. Higher values for Western countries (12–62%) were reported by Bont et al. [41]. The discrepancies between countries can be attributed to methodological differences.

In our study, RSV-associated hospitalization rates for children aged less than 5 years increased from 2.7 per 1000 population in the 2015/16 season to 11.9 per 1000 in the 2022/23 season, which is in line with data from other studies [11,42]. In contrast, Rząd et al. [40] included all RSV-related hospitalizations in Poland between 2010 and 2020, and the average annual admission rate over that time was 2.7 per 1000. However, this much lower value was calculated as the average of all seasons, mostly spanning the pre-COVID-19 period. Lower values reported by Rząd et al. [40] might thus result from including earlier non-COVID-19 pandemic seasons into analysis. Homaira et al. [10] reported a pre-pandemic RSV hospitalization rate of 4.9 per 1000 in Australia, which is in the range of rates observed with our study. 

Our data further confirm that younger age is an important risk factor for RSV-related hospitalization. The highest share of RSV-associated hospitalizations by age group was noted for infants aged 2–6 months, which is in line with other studies. Rząd et al. [40] reported that 81.7% of all RSV-coded hospitalizations in children aged less than 5 years in Poland occurred in infants aged less than 1 year. In other countries, the share of children younger than 1 year among RSV-related hospitalizations in the age cohort of 0–5 years ranged between 50% and 83.5% [10,11,37,43,44]. The incidence rate of hospitalizations for infants aged 0–12 months in our study was 18.1 per 1000 before the pandemic and 45.3 per 1000 after the pandemic. The pre-pandemic value is consistent with the values reported by other authors (8.6–35.1 per 1000) [23,27,45,46,47,48].

In our study, we found that the highest age-stratified rates of hospitalization were observed in infants aged 2–3 months, with a mean two-year rate of 94.8 per 1000 in the post-COVID period. This high rate persisted across various age groups in that time period, with infants aged 4–6 months experiencing a rate of 58.1 per 1000, those aged 0–1 month at 47 per 1000, and infants aged 7–12 months at 24.9 per 1000. Other authors also reported the highest RSV-associated hospitalization rates in children aged less than 12 months, although the differences between the individual age groups of infants were much lower [47,49]. This can be explained by the fact that in Poland, newborns have little social contact in the first weeks of life, which diminishes the risk of infection. Moreover, the discrepancies in hospitalization rates between various studies can be explained by differences in healthcare systems, coding and testing practices, RSV circulation, and methodology for the measurement of the incidence rate [19,50]. Overall, our data confirm that the highest burden of serious RSV disease is observed in children aged less than 1 year, and especially those younger than 6 months, as has been reported by other authors [10,11,19,20,43,47,51,52,53,54,55].

At the same time, our results showed an increasing share over time of older children in all RSV-coded hospitalizations, especially after the outbreak of the COVID-19 pandemic. The increase of pre-pandemic vs post-pandemic admission rates was much higher in children aged over 1 year (especially those aged >2 years) than in infants aged less than 1 year. Similar findings were reported by other authors [25,31,49,56]. This may be due to the increased population of RSV-naive children in older age groups because of public health measures for COVID-19 [22,24], i.e., the so-called ‘immunity debt’ theory. Another reason might be a detection bias, due to the increasing availability of cheaper and faster diagnostic tests after the pandemic, resulting in the growing rates of testing in all age groups, and especially in young children, with the highest increase (by 110%) noted in those aged 2–4 years [38]. 

In our study, most children hospitalized due to RSV infection (94.2%) had no known risk factors for severe RSV infection. This finding is in line with other studies reporting the share of children at no risk to range between 84% and 96.8% for France, Germany, Spain, Portugal, and Israel [37,42,43,45,47,57]. While the effect of comorbidities such as prematurity, lung disease, or congenital heart disease on the severity of RSV infection is well known, most hospitalizations occur in children with no such risk factors [50]. The low proportion of at-risk children in our population could also be due to the national program of RSV immunoprophylaxis, which has been in operation since 2008, covering most of the above high-risk groups [15]. Over 90% of eligible children received immunoprophylaxis in the 2021/22 season (data not published). Among children with congenital heart disease, bronchopulmonary dysplasia, and prematurity, palivizumab is associated with a 53%, 65%, and 68% reduction in RSV hospitalization, respectively [58]. In 2023, two products were approved for RSV prophylaxis: Nirsevimab, a long-acting monoclonal antibody intended for use in neonates and infants, and the maternal vaccine Abrysvo, intended for use during pregnancy to prevent severe illness in young infants. Both products require only one injection to provide protection for the duration of the typical RSV season, which can make RSV prophylaxis easier and more efficient, allowing for the protection of a wider group of children.

The major limitation of our study is that it is a retrospective analysis of real-world, standard of care diagnostic coding data rather than prospectively collected clinical data. Such coding data are not collected for research purposes can be affected by testing and coding practices. Without clinical data, potential coding errors cannot be identified, and the potential bias of testing mostly younger children cannot be assessed. This can lead to the omission of RSV-positive cases and the inclusion of RSV-negative ones [19,47]. Another limitation is the absence of information regarding the proportion of patients who received monoclonal antibodies for RSV prophylaxis, as it was not possible to obtain these data.

Our study has several strengths. Although we did not directly compare our use of coding data to another methodology, other studies have shown RSV-specific ICD-10 codes to be a good indicator to describe RSV epidemiology, although the number of cases may be underestimated [59]. As surveillance data strongly correlate with virologic test results in a pediatric population, RSV-coded admissions can be used as a proxy of RSV-confirmed admissions [19,59,60]. This type of data makes it possible to observe trends and patterns over time, which in turn supports the planning of healthcare services and provides data to inform and evaluate healthcare policy [19].

## 5. Conclusions

Most RSV-related hospitalizations in Polish children occur in young infants. The COVID-19 pandemic was associated with a change in the seasonal pattern of RSV outbreaks in terms of both time and intensity. Towards the end and after the pandemic, more RSV-related hospitalizations were observed in older children, aged 13 months and older, than were seen prior to the COVID-19 pandemic. Most RSV-associated hospitalized children had no known risk factors for RSV infection.

Variations in the seasonality of RSV infections are important for prevention planning, so it is crucial to continue tracking seasonal patterns. The fact that the majority of children hospitalized with RSV were otherwise healthy and without known risk factors suggests that prevention should be extended to a broader group, especially the youngest children below 1 year of age. Wide access to diagnostic tests to accurately assess the extent of the disease burden is also important.

## Figures and Tables

**Figure 1 viruses-16-00704-f001:**
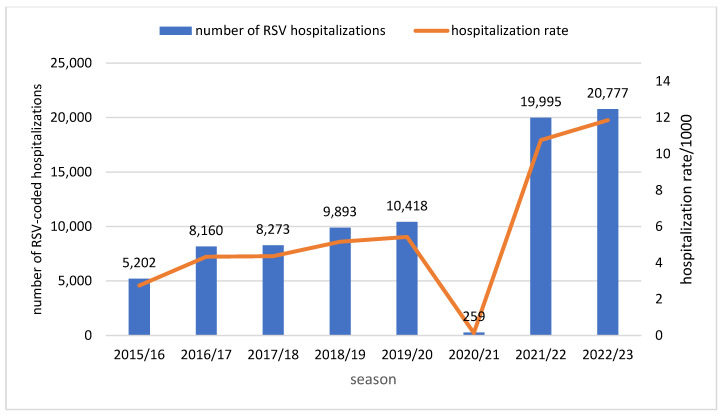
Number of RSV-related hospitalizations and hospitalization rates per 1000 in children aged <5 years, by season in 8 consecutive seasons, Poland.

**Figure 2 viruses-16-00704-f002:**
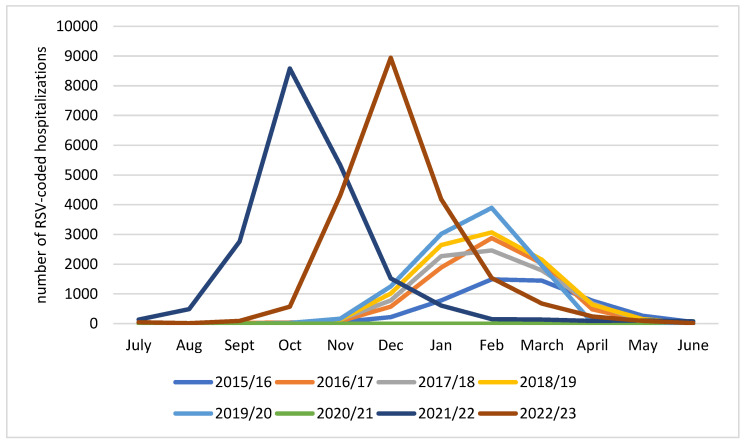
Number of hospitalizations per month in 8 consecutive seasons in children aged <5 years, Poland.

**Figure 3 viruses-16-00704-f003:**
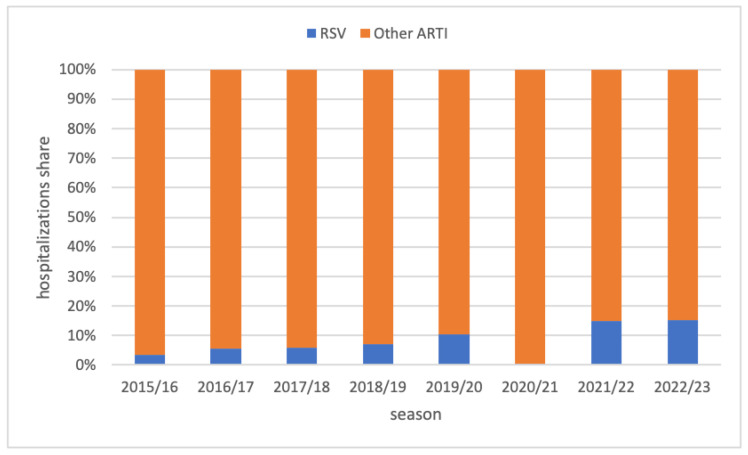
Proportion of RSV-related hospitalizations among all ARTI hospitalizations in 8 consecutive seasons in children aged <5 years, Poland.

**Figure 4 viruses-16-00704-f004:**
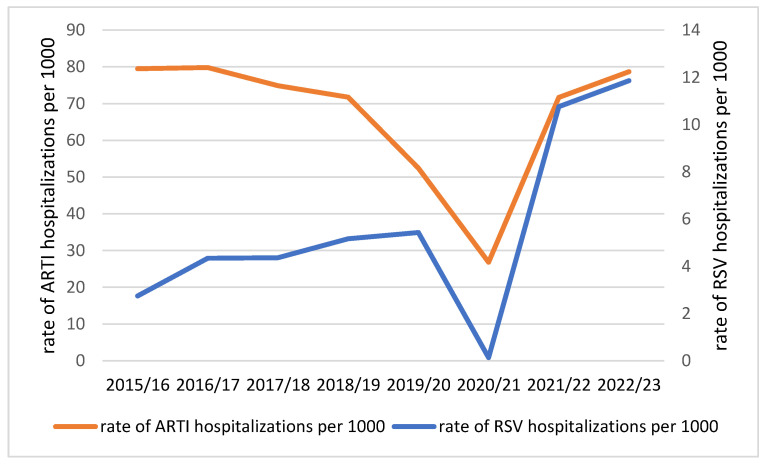
Rates of RSV- and ARTI-related hospitalizations in 8 consecutive seasons in children aged <5 years, Poland.

**Figure 5 viruses-16-00704-f005:**
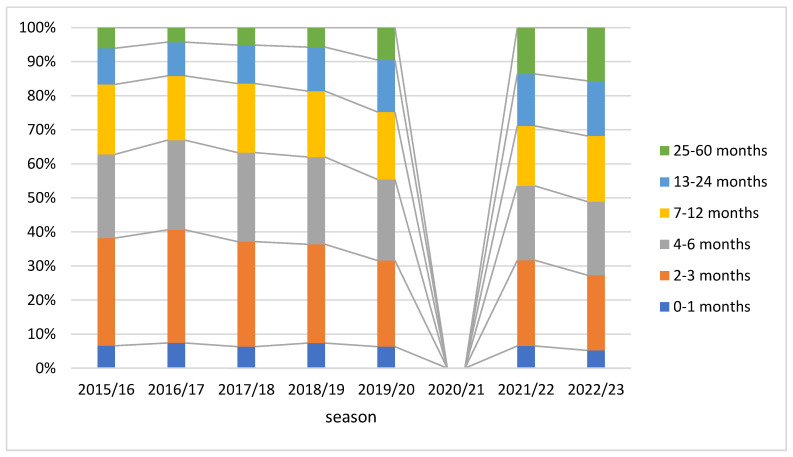
Impact of age groups on the total number of RSV-related hospitalizations by age group across seasons, children < 5 years old, Poland.

**Table 1 viruses-16-00704-t001:** Share of children in high-risk vs no-risk groups in all RSV-coded hospitalizations; % (n).

Period	Age Group, Years			
	0–12		13–60	
	High-Risk Group	No-Risk Group	High-Risk Group	No-Risk Group
Pre-COVID-19	5.3% (1793)	94.7% (32,342)	7.7% (601)	92.3% (7210)
Post-COVID-19	5.3% (1508)	94.7% (26,882)	7.3% (901)	92.7% (11,481)

**Table 2 viruses-16-00704-t002:** RSV-associated hospitalization rates per 1000 for all age groups and overall, across seasons.

Season		Age Group (Months)				*p*-Value
		0–12	13–24	25–60	0–60	
Pre-COVID-19	2015/16	12.0	1.5	0.3	2.7	
	2016/17	18.7	2.2	0.3	4.3	
	2017/18	17.5	2.4	0.4	4.4	
	2018/19	21.2	3.2	0.5	5.2	
	2019/20	21.4	4.1	0.9	5.4	
	Mean (SD)	**18.1 (3.8)**	**2.7 (1.0)**	**0.5 (0.2)**	**4.4 (1.1)**	**<0.001**
Post-COVID-19	2021/22	43.3	8.7	2.3	10.8	
	2022/23	47.3	10.0	2.9	11.9	
	Mean (SD)	**45.3 (2.8)**	**9.4 (0.9)**	**2.6 (0.4)**	**11.3 (0.8)**	**<0.001**
Post-/pre-COVID-19 ratio		**2.5**	**3.5**	**5.6**	**2.6**	
*p*-Value			**<0.001**			

SD—standard deviation.

**Table 3 viruses-16-00704-t003:** RSV-associated hospitalization rates per 1000 children, by age groups below 1 year, and across seasons.

Season		Age Group (Months)				*p*-Value
		0–1	2–3	4–6	7–12	
Pre-COVID-19	2015/16	11.0	26.4	13.7	5.7	
	2016/17	18.3	41.0	21.7	7.8	
	2017/18	15.8	38.8	21.9	8.5	
	2018/19	23.3	45.7	26.9	10.1	
	2019/20	21.5	42.6	26.8	11.2	
	Mean (SD)	**18.0 (4.9)**	**38.9 (7.4)**	**22.2 (5.4)**	**8.7 (2.1)**	**<0.001**
Post-COVID-19	2021/22	49.1	94.6	54.6	22.1	
	2022/23	44.9	94.9	61.6	27.6	
	Mean (SD)	**47.0 (3.0)**	**94.8 (0.2)**	**58.1 (4.9)**	**24.9 (3.9)**	**0.001**
Post-/pre-COVID-19 ratio		**2.6**	**2.4**	**2.6**	**2.9**	
*p*-value			***p* < 0.001**			

SD—standard deviation.

## Data Availability

Data are contained within the article.

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
