# Peer review of "Epidemiology of Respiratory Syncytial Virus Hospitalizations in Poland: An Analysis from 2015 to 2023 Covering the Entire Polish Population of Children Aged under Five Years"

_viruses, 2024, doi:10.3390/v16050704_

Round 1
Reviewer 1 Report
Comments and Suggestions for Authors
Estimated Authors,
I've read with great interest the present article on the RSV hospitalizations in Poland. The study, based on health registries, is both well written and consistent with international data on RSV epidemiology. In fact, Authors have omitted from their analyses one RSV season (i.e. pandemic one) for reasons that quite reasonable. In fact, they reported, discussed and tentatively shared some likely explanation about the causes of the substantial increase of hospitalization rates after pandemic season 2019/2020.
Honestly, I've neither concerns nor doubts about the design and the outline of this study. Only two minor issues should be fixed before the acceptance:
1) please report crude numbers and not only percent values in table 1;
2) Table 2 and Table 3, please include 95%CI for all estimates;
3) Please report the source of demographic data for the reference population used for calculating hospitalization data.
Good job!
Author Response
I've read with great interest the present article on the RSV hospitalizations in Poland. The study, based on health registries, is both well written and consistent with international data on RSV epidemiology. In fact, Authors have omitted from their analyses one RSV season (i.e. pandemic one) for reasons that quite reasonable. In fact, they reported, discussed and tentatively shared some likely explanation about the causes of the substantial increase of hospitalization rates after pandemic season 2019/2020.
Honestly, I've neither concerns nor doubts about the design and the outline of this study. Only two minor issues should be fixed before the acceptance:
Authors: Thank you for this review. Please find below our point-by-point answers.
- please report crude numbers and not only percent values in table 1;
Authors: We have provided crude numbers.
- Table 2 and Table 3, please include 95%CI for all estimates;
Authors: Thank you for this suggestion. After consulting with a statistician, we decided to add the standard deviation because our means resulted from too small sample.
- Please report the source of demographic data for the reference population used for calculating hospitalization data.
Authors: We have added explanation that reference data came from Polish Central Statistical Office.
Good job!
Authors: Thank you very much
Reviewer 2 Report
Comments and Suggestions for Authors
Academic Critique of "Epidemiology of Respiratory Syncytial Virus Hospitalizations in Poland: an Analysis from 2015 to 2023 Covering the Entire Polish Population of Children Aged under Five Years"
Introduction and Background: The paper investigates the epidemiology of Respiratory Syncytial Virus (RSV) hospitalizations in Poland among children under five years old from 2015 to 2023, covering eight consecutive seasons. It aims to estimate the rates of RSV-related hospitalizations, especially concerning the impact of the COVID-19 pandemic on RSV seasonality and hospitalization patterns. The introduction effectively contextualizes the importance of RSV infections, its seasonality, and its global burden, providing a clear rationale for the study.
Methods: The study employs a retrospective observational cohort design utilizing data obtained from the National Health Fund in Poland. It includes detailed definitions of age groups, diagnostic codes, and data analysis methods. The approach is robust, utilizing established coding systems and appropriate statistical techniques for analyzing seasonal trends and age-specific hospitalization rates.
Results: The results section presents comprehensive findings regarding RSV-related hospitalizations, including temporal trends, age-specific patterns, and the impact of the COVID-19 pandemic. The presentation of results is clear, with well-designed figures and tables facilitating understanding. The study effectively highlights the significant increase in RSV-related hospitalizations following the COVID-19 pandemic and the shift in seasonality observed during the pandemic period.
Discussion: The discussion section critically interprets the findings in the context of existing literature, addressing potential explanations for observed patterns and implications for public health. It discusses the disruption of RSV seasonality by the COVID-19 pandemic, the increased hospitalization rates, and the changing age distribution of RSV cases. However, some aspects, such as the potential impact of testing practices on hospitalization rates, could be further elaborated.
Strengths and Limitations: The study's strengths lie in its comprehensive analysis of a large dataset covering multiple seasons and its contribution to understanding RSV epidemiology in Poland. However, limitations, such as the reliance on coding data without clinical validation and potential biases in testing and coding practices, are acknowledged. These limitations are crucial for interpreting the results accurately and understanding their implications.
Conclusion: Overall, the paper provides valuable insights into the epidemiology of RSV-related hospitalizations in Poland, especially regarding the impact of the COVID-19 pandemic. It effectively combines rigorous methodology, comprehensive data analysis, and critical interpretation of findings. However, acknowledging limitations and discussing potential biases strengthen the paper's credibility and provide avenues for future research.
Main Question Addressed by the Research:
The main question addressed by the research is the epidemiology of Respiratory Syncytial Virus (RSV) hospitalizations in Poland from 2015 to 2023. Specifically, the study aims to investigate the burden of RSV-related hospitalizations, including age distribution, seasonal patterns, and the impact of risk factors such as prematurity, chronic lung disease, and congenital heart disease.
Originality and Relevance to the Field:
The study's originality lies in its focus on the epidemiology of RSV hospitalizations in Poland over a nine-year period. This research is relevant as it contributes to the understanding of RSV burden, particularly in a specific geographic region, aiding in the development of targeted prevention and management strategies. The paper addresses a gap in the field by providing comprehensive data on RSV hospitalizations in Poland, which may not have been extensively studied previously.
Contribution to the Subject Area:
The paper adds valuable data on RSV hospitalizations in Poland, which can be compared with studies from other regions to identify similarities and differences in RSV epidemiology. By providing detailed information on age distribution, seasonal trends, and risk factors, the study contributes to the existing knowledge base on RSV epidemiology, potentially informing healthcare policies and interventions.
Methodological Improvements and Further Controls:
The authors should consider improving the methodology by including more robust statistical analyses, such as multivariate regression models to adjust for potential confounding factors. Additionally, further controls could involve stratifying the analysis by geographic region within Poland to account for potential variations in RSV epidemiology.
Consistency of Conclusions with Evidence:
The conclusions drawn in the paper appear consistent with the evidence and arguments presented. The study effectively addresses the main questions posed by analyzing RSV hospitalization data over the specified time period and examining relevant risk factors. However, further experiments or analyses may be needed to explore any discrepancies or unanswered questions that arise from the research.
Appropriateness of References:
The bibliography provided seems appropriate for a research paper on respiratory syncytial virus (RSV) infection and its associated burden, epidemiology, prevention, and treatment strategies. It includes a wide range of sources such as peer-reviewed journal articles, conference abstracts, government reports, and epidemiological studies. These references cover various aspects of RSV infection, including its global burden, transmission, clinical manifestations, risk factors, hospitalization trends, seasonal patterns, and the impact of interventions and public health measures.
Comments on the Quality of English Language
Minor English changes must be done.
Author Response
Authors: Thank you for such great review. According to this suggestion: The authors should consider improving the methodology by including more robust statistical analyses, such as multivariate regression models to adjust for potential confounding factors. We introduced statistics which enable groups comparisons. Unfortunately, we cannot provide analysis by geographic regions because Poland is centralized in terms of the system's organization, the role of the payer/controller, and infection reporting. Hence, our decision to write about the entire population of Poland. However, it is a great idea to observe the change over time and further analyze the data once the approach is standardized across regions.
Additionally, we have added Figure 4, which describes the rates of RSV- and ARTI-related hospitalizations. This figure confirms the impact of testing practices on hospitalization rates, as mentioned in the discussion.
Reviewer 3 Report
Comments and Suggestions for Authors
I was invited to revise the paper entitled "Epidemiology of Respiratory Syncytial Virus Hospitalizations in Poland: an Analysis from 2015 to 2023 Covering the Entire Polish Population of Children Aged under Five Years". It was a retrospective study aimed to evaluate the hospitalization rate for severe RSV in Poland from 2015 to 2023. Authors extracted data from hospital records reporting RSV ICD codes.
The topic is interesting and no previous studies were performed in Poland on RSV admissions in my knowledge.
Major observations:
- The paper should be enriched in word count;
- Introduction section was poor and shold be improved;
- About methods, Authors should standardize the incidence rate by age and gender based on Poland residend population;
- It is important to highlight patients' outcomes. How many death? How many ICU admissions? Lenght of stay?
- Trend should be tested with a specific statistical model;
- It is important to evaluate also if patients aged less than 1y were born preterm. If so, did they take monoclonal antibodies (pavilizumab)?
- Recent studies performed also in Italy reported similar results.
- Authors should discuss tnew preventive strategies that can be used as public health intervention such as vaccine and new long acting monoclonal antibody;
Author Response
I was invited to revise the paper entitled "Epidemiology of Respiratory Syncytial Virus Hospitalizations in Poland: an Analysis from 2015 to 2023 Covering the Entire Polish Population of Children Aged under Five Years". It was a retrospective study aimed to evaluate the hospitalization rate for severe RSV in Poland from 2015 to 2023. Authors extracted data from hospital records reporting RSV ICD codes.
The topic is interesting and no previous studies were performed in Poland on RSV admissions in my knowledge.
Authors: Thank you for this review. Please find below our point-by-point answers.
Major observations:
- The paper should be enriched in word count;
Authors: We have enriched word count in our manuscript.
- Introduction section was poor and shold be improved;
Authors: Thank you for this review. We have revised and improved introduction section.
- About methods, Authors should standardize the incidence rate by age and gender based on Poland residend population;
Authors: We have included data on the proportion of male live births. We presumed that standardizing the incidence rate by age was unnecessary, as the numbers of men and women are comparable, and the proportions remain relatively stable over the analyzed period.
- It is important to highlight patients' outcomes. How many death? How many ICU admissions? Lenght of stay?
Authors: We have supplied data regarding the number of hospitalizations that resulted in death. However, data on hospitalizations involving procedures in the ICU are reserved for a separate publication and are therefore not included here.
- Trend should be tested with a specific statistical model;
Authors: We have provided comparisons between age groups and for post-/pre-COVID-19 ratios.
- It is important to evaluate also if patients aged less than 1y were born preterm. If so, did they take monoclonal antibodies (pavilizumab)?
Authors: Our study was based on data obtained from the National Health Found in Poland. They lack immunization data, which we are unable to obtain. Additionally, data on risk factors may be prone to reporting bias due to funding issues.
- Recent studies performed also in Italy reported similar results.
- Authors should discuss tnew preventive strategies that can be used as public health intervention such as vaccine and new long acting monoclonal antibody;
Authors: Thank you for this suggestion. We have added a paragraph on preventive strategies to the discussion.
Reviewer 4 Report
Comments and Suggestions for Authors
The submitted article addresses a topic of interest, however it suffers from critical issues that must be resolved before evaluating its publication.
1) How were the data obtained? There is no indication relating to the authorization of an Ethics Committee, nor is there any mention of informed consent in compliance with current legislation.
2) The data presented should be compared in terms of significance after applying statistical tests.
3) In several parts of the text the overall increase in hospitalizations in older children due to RSV infection is mentioned, but the causes are poorly addressed (e.g. more virulent circulating strains, absence of prophylaxis, etc.).
4) In figure 2 the months must be reported with their denomination, as it is difficult to follow the relative description.
5) The impact of this study and future projections must be better highlighted in the conclusions.
Author Response
The submitted article addresses a topic of interest, however it suffers from critical issues that must be resolved before evaluating its publication.
- How were the data obtained? There is no indication relating to the authorization of an Ethics Committee, nor is there any mention of informed consent in compliance with current legislation.
Authors: The data were obtained from the National Health Fund and the Polish Central Statistical Office, as mentioned in sections 2.1 and 2.3. Since we used anonymized data that is freely available in the public domain, we believe that there is no need for Ethics Committee approval or informed consent.
- The data presented should be compared in terms of significance after applying statistical tests.
Authors: Thank you for this suggestion, we have added statistical comparisons.
- In several parts of the text the overall increase in hospitalizations in older children due to RSV infection is mentioned, but the causes are poorly addressed (e.g. more virulent circulating strains, absence of prophylaxis, etc.).
Authors: We identified two major factors contributing to increased hospitalization rates: the expiration of regulatory measures recommending the use of personal protective equipment during the COVID-19 pandemic, and the limited operational mode of the system, along with reduced social contacts. Additionally, growing rates of testing across all age groups also played a role. These factors are supported by our data and are discussed in the discussion section.
- In figure 2 the months must be reported with their denomination, as it is difficult to follow the relative description.
Authors: Thank you for this suggestion, we have modified Figure 2.
- The impact of this study and future projections must be better highlighted in the conclusions.
Authors: Thank you for this suggestion, we have improved conclusions section.
Round 2
Reviewer 3 Report
Comments and Suggestions for Authors
Authors addressed the great part of my previous comments. I suggest to add the lack in monoclonal antibobies information among limitations.
Author Response
Thank you, we have added information on monoclonal antibodies to the limitations as suggested.
Reviewer 4 Report
Comments and Suggestions for Authors
The authors have sufficiently adressed the criticisms highlighted.
Author Response
Thank you.